# Diversity of the *Cnaphalocrocis medinalis* gut bacterial community and its contribution to reproduction

Zhichao Yao,[1] Chengrui Xue,[1] Yang Ang,[1] Ziwei Wu,[1] Xin Liu,[1] Ying Liu,[1] Fang Liu,[1] Qinjian Pan[2,3]

**ABSTRACT** The pest *Cnaphalocrocis medinalis* has a high reproductive rate and causes significant damage in rice. Previous research demonstrated the effect of larval gut microbiota on the fecundity of *C. medinalis*, but the role of adult gut bacteria in reproduction remains unclear. Here, we characterized the gut microbiota of field and laboratory adult populations using 16S rRNA sequencing and examined their effects on female reproduction. *Enterococcus*, *Asaia*, *Enterobacter*, and unclassified Enterobacteriaceae constituted 55%–85% of the microbiota in laboratory adults, while *Asaia*, *Acinetobacter*, *Apibacter*, unclassified Orbaceae, and unclassified Enterobacteriaceae comprised ~50% of the microbiota in field adults. Several genera, including *Enterobacter*, were shared between the two populations. The abundance of *Enterobacter* increased gradually during ovarian development in laboratory females and was enriched in field female guts, showing a positive correlation with reproductive activity. Antibiotic-mediated microbiota depletion significantly reduced ovariole length, ovarian size, and egg deposition in females fed with a 2.5% honey solution (HS) diet but not with 5% or 10% HS. A total of 17 and 14 bacterial species were isolated from laboratory and field adults, respectively. Monoassociation with *Enterobacter bugandensis* (from laboratory females) or *Enterobacter roggenkampii* (from field females) increased egg production by 1.73- and 1.94-fold as compared to antibiotic-treated females supplied with 2.5% HS, respectively. Monoassociation with *Providencia rettgeri* from laboratory females reduced egg production by 51.17% as compared to the conventionally reared females supplied with 10% HS. These results indicate that gut microbiota regulate reproduction in *C. medinalis* and present a potential target for sustainable pest management.

**IMPORTANCE** The rice leaf roller, *Cnaphalocrocis medinalis*, is a notorious migratory rice pest whose high reproductive rate drives population growth. However, the role of its gut microbiota in regulating reproduction remains poorly understood. Here, we demonstrate that gut bacteria are essential for the fecundity of *C. medinalis* under nutrient stress. Depleting gut bacteria severely impaired reproduction, while reintroducing two key symbionts, *Enterobacter bugandensis* and *Enterobacter roggenkampii* (isolated from laboratory and field *C. medinalis* adult populations, respectively), significantly restored reproductive capacity under low-nutrient conditions. These bacteria were predominantly located in the female adult gut. In contrast, monoassociation with *Providencia rettgeri* substantially reduced fecundity even under normal nutrient conditions. Our work highlights the contribution of gut bacteria in lepidopteran insect reproduction and provides novel insights into gut microbe-host symbiosis as well as host nutritional adaptation.

**KEYWORDS** *Cnaphalocrocis medinalis*, gut bacteria, temporal change, reproduction, *Enterobacter bugandensis*, *Enterobacter roggenkampii*

**Peer Reviewer** Letian Xu, State Key Laboratory of Biocatalysis and Enzyme Engineering, Wuhan, China

Address correspondence to Fang Liu, liufang@yzu.edu.cn, or Qinjian Pan, qjpan@yzu.edu.cn.

The authors declare no conflict of interest.

The insect gut harbors a diverse microbiota that modulates host development and physiology. This microbial community enhances host fitness by providing essential nutrients (1), regulating nutritional and metabolic homeostasis (2, 3), and promoting growth (4). Due to these multifaceted contributions, the gut microbiome is considered to be an "organ within an organ."

The gut microbiomes of over a hundred lepidopteran species have been surveyed, but it is unclear whether lepidopteran larvae host stable, functional microbial communities (5, 6). For example, Hammer et al. (5) found no evidence of host-specific resident symbionts in 124 species of wild, leaf-feeding caterpillars; furthermore, the elimination of gut bacteria in *Manduca sexta* had no significant impact on development, weight gain, or survival. Conversely, recent studies indicate that some lepidopteran larvae, such as *Tyria jacobaeae* and *Mamestra brassicae*, harbor stable, resident gut microbiomes (7, 8). Moreover, specific gut symbionts have been linked to functional roles; for example, *Pseudomonas fulva* ZJU1 aids *Bombyx mori* in digesting plant toxins (9), and the gut microbiota in *Hyphantria cunea* helps the insect adapt to new host plants (10). Multiple studies have focused on the microbiome in lepidopteran larvae, but the functional roles of the gut microbiome in lepidopteran adults remain limited.

Gut microorganisms have crucial roles in regulating the reproduction of many insects. For instance, removal of gut symbiotic microbes significantly reduced the reproductive capacity of *Drosophila melanogaster*, *Spodoptera frugiperda*, and *Bactrocera dorsalis* (11–13). However, the reintroduction of specific symbionts, such as *Acetobacter* spp. in *D. melanogaster* and *Enterobacter hormaechei* in *B. dorsalis*, restored host fertility (3, 13). Despite extensive research, the role of gut microbes in regulating lepidopteran reproduction remains elusive.

The rice leaf folder, *Cnaphalocrocis medinalis* (Lepidoptera: Pyralidae), is a destructive, migratory pest of rice in Asia. During feeding, larvae scrape away healthy leaf tissue within folded leaves, thus reducing photosynthetic activity and yields (14). Chemical pesticides effectively control *C. medinalis* larvae, but their use has resulted in resistance, pollution, and harm to non-target organisms (15); this has spurred interest in sustainable alternatives, such as gut symbiont-based strategies for targeted pest control (16, 17). Multiple studies have characterized the gut bacterial communities in *C. medinalis* and their function. These include analyses of the diversity and abundance of gut bacteria in various life stages (18), the impact of geography, host plants, and baculovirus infection on larval gut microbiota (19–21), the contribution of gut microbiota to larval nutrition (22), and the effects of gut bacteria on *C. medinalis* fitness (23). Unfortunately, the functional roles of the gut microbiome in regulating adult reproduction remain unclear.

In this study, the diversity and composition of gut bacterial communities were characterized in laboratory and field populations of *C. medinalis* adults, and the bacterial genera shared between the two populations, as well as changes in their abundance across four age groups of laboratory adults and between field female and male, were investigated. Associations between gut microbiota and female fecundity were also investigated in *C. medinalis*. Using monoassociated gnotobiotic assays, the contribution of culturable gut bacteria from both laboratory and field populations was assessed for potential roles in *C. medinalis* reproduction. These findings advance the understanding of gut microbial functions in *C. medinalis* reproduction and offer a hypothetical basis for developing new pest control strategies that target the gut microbiota and reproduction of insects.

## RESULTS

### Changes in gut bacterial load during ovarian development in a laboratory population of *C. medinalis* female adults

To determine whether the gut microbiota influences reproduction in *C. medinalis*, ovarian development and its correlation with fecundity were examined. From 0 to 2 days post-emergence (DPE), ovariole lengths increased from $5.40 \pm 0.23$ mm to $9.27 \pm 0.29$ mm ($P < 0.01$), and ovarian size increased from $4.81 \pm 0.21$ mm$^2$ to $13.68 \pm 0.62$ mm$^2$

($P < 0.01$); values peaked between 2 and 4 DPE and began to atrophy at 5 DPE (Fig. 1A through C). This developmental pattern was also reflected in the grading of ovaries. The proportion of mature ovarioles (stages III–IV) increased from 0% to 55.56% between 0 and 2 DPE and exceeded 90% during 3–4 DPE, while 41.67% reached stage V by 5 DPE (Fig. 1D). The oviposition profile showed that females began laying approximately 20 eggs/day at 3 DPE, with peak spawning (~45 eggs/day) occurring between 4 and 6 DPE (Fig. 1E). These findings outline the dynamics of post-eclosion ovarian development and oviposition.

Based on the indicators of ovarian development and fecundity in Fig. 1A through E, the following four age groups were selected for analysis of gut bacterial loads: 0 DPE, pre-feeding; 1 DPE, post-feeding; 3 DPE, onset of oviposition; and 5 DPE, peak

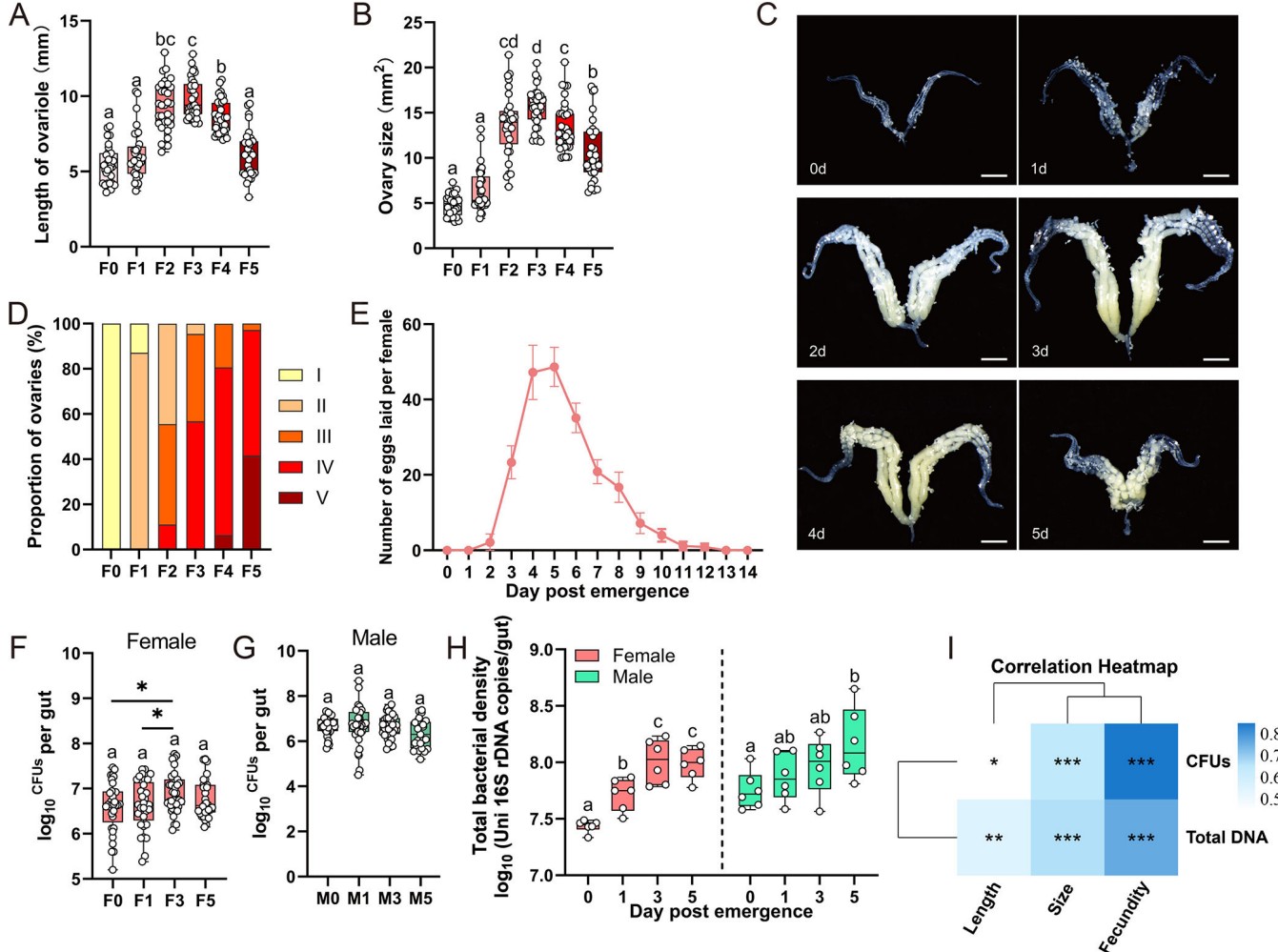

FIG 1 Temporal changes in gut bacterial load of *C. medinalis* adults. (A) Temporal changes in ovariole length and (B) ovarian size in *C. medinalis* female adults at 0 (F0) to 5 (F5) days post-emergence. The lower and upper limits of rectangles define the 25th and 75th percentiles, and median values are represented by black lines. The open circles represent biological replicates in panels A ($n = 30$–36) and B ($n = 30$–36). (C) Representative images of ovaries at 0–5 days post-emergence. Scale bars = 1 mm. (D) The percentage of ovaries in females at the following developmental stages: I, transparent opalescent phase; II, yolk deposition; III, egg maturation; IV, peak oviposition phase; and V, a terminal phase. (E) Number of eggs laid by females fed with a 10% honey solution and monitored for emergence from 0 to 14 days. Bacterial colony-forming units (CFUs) in the gut of (F) females ($n = 33$–39) and (G) males ($n = 25$–36) at 0, 1, 3, and 5 days post-emergence. *$P < 0.05$ with a two-tailed Student's *t*-test. (H) Total bacterial density as assessed by qPCR ($n = 6$) in the whole gut of females and males at 0, 1, 3, and 5 days post-emergence. (I) Spearman correlation analysis of bacterial density (qPCR and $\log_{10}$CFU) and reproductive indices. *$P < 0.05$, **$P < 0.01$, and ***$P < 0.001$ with R statistical analysis. Multiple comparisons in panels A, B, F, G, and H were performed using one-way analysis of variance with Tukey's post hoc test. Different letters indicate significant differences at $P < 0.05$. Abbreviations: F0, F1, F3, and F5 represent females at 0, 1, 3, and 5 days post-emergence; M0, M1, M3, and M5 represent males at 0, 1, 3, and 5 days post-emergence.

spawning. Consistent with ovarian development and host fecundity, the number of bacterial colony-forming units (CFUs) and bacterial density in females significantly increased 1.75-fold ($P < 0.05$) and 4.04-fold ($P < 0.01$), respectively, from 0 to 3 DPE (Fig. 1F and H). In contrast, these parameters remained stable in males over the same period (Fig. 1G and H). Spearman correlation analysis further revealed positive correlations between temporal changes in gut bacterial load in females and ovarian development and fecundity (Fig. 1I). These results indicated that the gut bacterial load in female *C. medinalis* exhibited temporal dynamics that coincided with reproductive activity, suggesting a potential role for gut bacteria in mediating female reproduction.

## Gut bacterial diversity and composition in a laboratory population of *C. medinalis* adults and its correlation with female reproduction

The connection between gut microbiota and *C. medinalis* reproduction was further explored by studying the composition of gut bacterial communities in four different adult age groups using Illumina MiSeq sequencing. Species accumulation curves indicated sufficient sampling depth and approached saturation (Fig. S1A). Newly emerged males harbored the largest quantity of observed species and the greatest richness when compared with other time points in males (Fig. 2A). At 0 DPE, females and males both exhibited the lowest evenness and were dominated by a few bacterial species, as reflected by the Shannon and Simpson diversity indices (Fig. 2A). Beta-diversity analysis showed the existence of significant temporal variations in the bacterial community structure across both sexes. PCO1 and PCO2, the first two principal coordinates, explained 61.52% and 11.95% of the variation, respectively (Fig. 2B). Multifactorial permutational multivariate analysis of variance (PERMANOVA) analysis revealed that age ($R^2 = 0.776$, $P < 0.001$), rather than sex ($R^2 = 0.018$, $P = 0.097$), was the primary driver of differences in community structure. These results indicated pronounced age-related temporal changes in the structure of gut microbiota during ovarian development.

Taxonomic analysis showed that the gut microbiota of all samples was dominated by Proteobacteria and Firmicutes, which collectively represented over 95% of the community (Fig. S1B). At the genus level, *Enterococcus*, *Asaia*, *Enterobacter*, and unclassified Enterobacteriaceae constituted approximately 55%–85% of the microbiota in both sexes (Fig. 2C). Venn analysis identified 23 and 31 bacterial genera shared across all four age groups in females and males, respectively, and most of the shared genera were *Enterococcus*, *Asaia*, or *Enterobacter*, confirming their status as core taxa in the *C. medinalis* gut (Fig. 2D). Interestingly, the relative abundances of the core taxa shifted with time (Fig. 2E through H). *Enterococcus* decreased significantly with age in both sexes (Fig. 2E), whereas *Asaia* increased with age (Fig. 2F). Unlike males, *Enterobacter* and unclassified Enterobacteriaceae in females increased significantly (2.12- and 2.08-fold, respectively; $P < 0.05$) at 1 DPE as compared to 0 DPE and then stabilized (Fig. 2G and H). Linear discriminant analysis effect size (LEfSe) analysis showed that temporal changes in these core taxa were the main drivers of community differences (Fig. S1C). Furthermore, Spearman correlation analysis revealed positive correlations between the relative abundances of *Enterobacter*, unclassified Enterobacteriaceae, *Asaia*, *Providencia*, and *Gluconobacter* and the host ovarian developmental index and fecundity, while *Enterococcus* and *Serratia* were negatively correlated (Fig. 2I). Collectively, these results suggest a close association between the core gut microbiota and reproduction in *C. medinalis*.

## Gut bacterial diversity of field *C. medinalis* adults and comparative analysis with laboratory populations

The gut microbiota in a field population of *C. medinalis* was analyzed to compare the similarities and differences in the composition of gut microorganisms between laboratory and field populations. Since accurately determining the developmental stages of adults collected in the field is challenging, this study focused exclusively on adults at the mature stage. Male adults exhibited significantly higher microbial diversity than females,

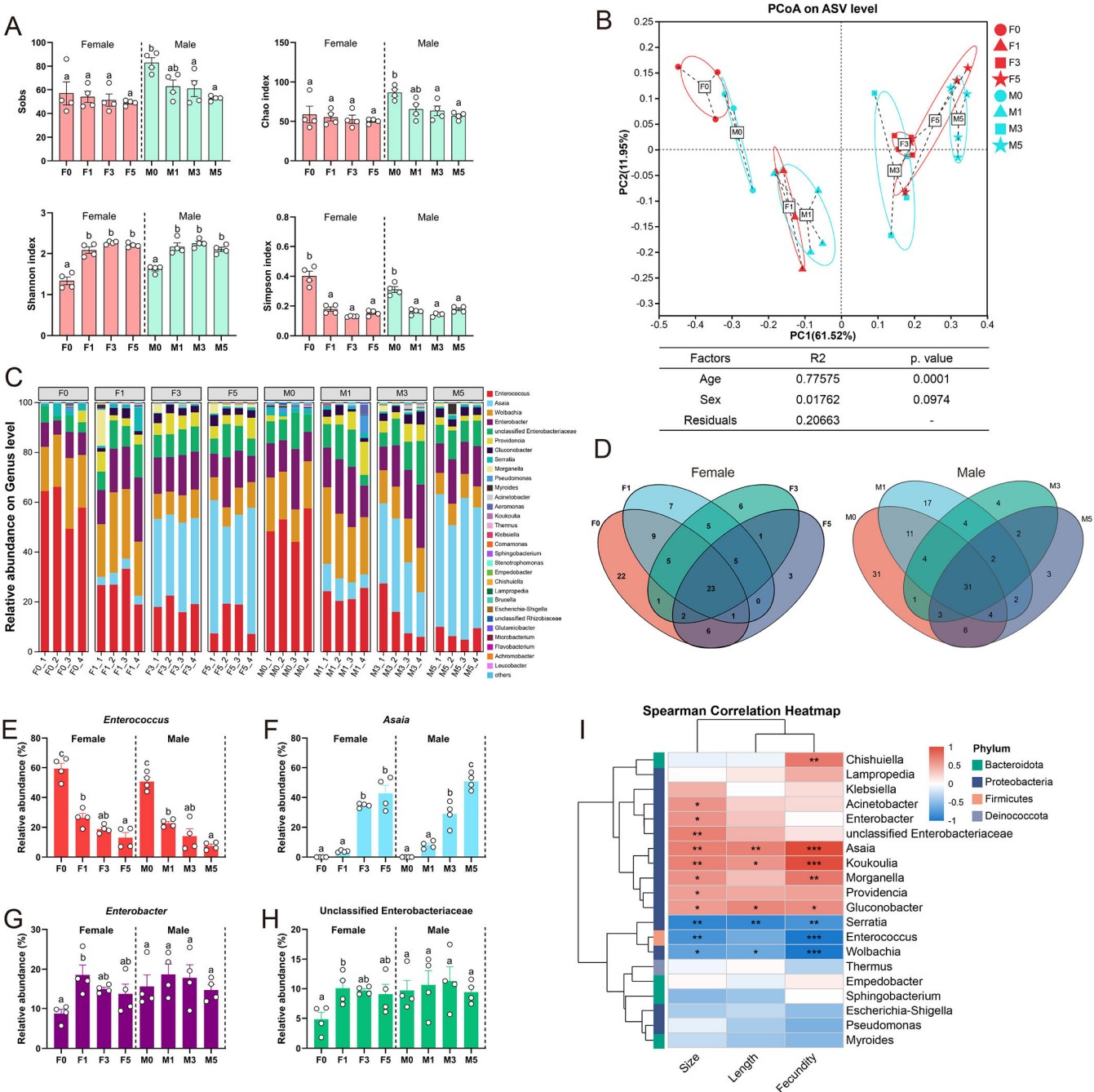

**FIG 2** Temporal changes in gut bacterial diversity and composition in *C. medinalis* adults. (A) Alpha-diversity indices (Sobs, Chao1, Shannon, and Simpson) of the gut bacterial community in four age groups of female and male adults (*n* = 4). (B) Principal coordinate analysis (PCoA) of bacterial communities based on Bray-Curtis dissimilarities. Community differences were assessed by PERMANOVA (Adonis test with 999 permutations). Red and blue represent female adults and male adults, respectively, and different ages are distinguished by different shapes. (C) Stacked bar plots showing the relative abundance of bacterial genera in four adult age groups (*n* = 4). (D) Venn diagrams of shared genera across different age groups in females and males. (E–H) Temporal changes in the relative abundance of four major genera, including (E) *Enterococcus*, (F) *Asaia*, (G) *Enterobacter*, and (H) unclassified Enterobacteriaceae. (I) Spearman correlation heatmap of the top 20 bacterial genera and their reproductive indices. Red and blue hues denote positive and negative correlations, respectively. *P < 0.05, **P < 0.01, ***P < 0.001. Multiple comparisons in panels A and E–H were performed separately for male and female groups using one-way analysis of variance followed by Tukey's post hoc test; different letters indicate significance at *P* < 0.05. Abbreviations: F0, F1, F3, and F5 represent females at 0, 1, 3, and 5 days post-emergence; M0, M1, M3, and M5 represent males at 0, 1, 3, and 5 days post-emergence.

whereas no significant differences in microbial richness were observed between sexes (Fig. 3A). Principal coordinate analysis (PCoA) revealed a significant separation in bacterial community similarity among females and males (Fig. 3B; PERMANOVA: $R^2 = 1$, $P = 0.034$). The gut microbiota of both sexes was dominated by Proteobacteria, Bacteroidota, and Firmicutes, which collectively accounted for over 99% of the community (Fig. S1B). The most abundant genera were *Acinetobacter*, *Asaia*, *Apibacter*, unclassified Enterobacteriaceae, and unclassified Orbaceae and collectively represented approximately 50% of the microbiota in both sexes (Fig. 3C), which is a profile distinct from that of the laboratory population.

Venn diagram analysis comparing gut microbiota between field females and laboratory females at 3 DPE identified 28 shared bacterial genera, dominated by *Asaia*, *Acinetobacter*, *Enterococcus*, unclassified Enterobacteriaceae, *Enterobacter*, and *Providencia* (Fig. 3D through I). The relative abundance of *Acinetobacter* and *Enterobacter* was significantly higher in field females than in field males (Fig. 3F and H), whereas an opposite pattern was observed for *Providencia* (Fig. 3I). Because *Enterobacter* showed more stable colonization than *Acinetobacter* across both populations, we further quantified its absolute abundance. qPCR analysis revealed that the absolute abundance

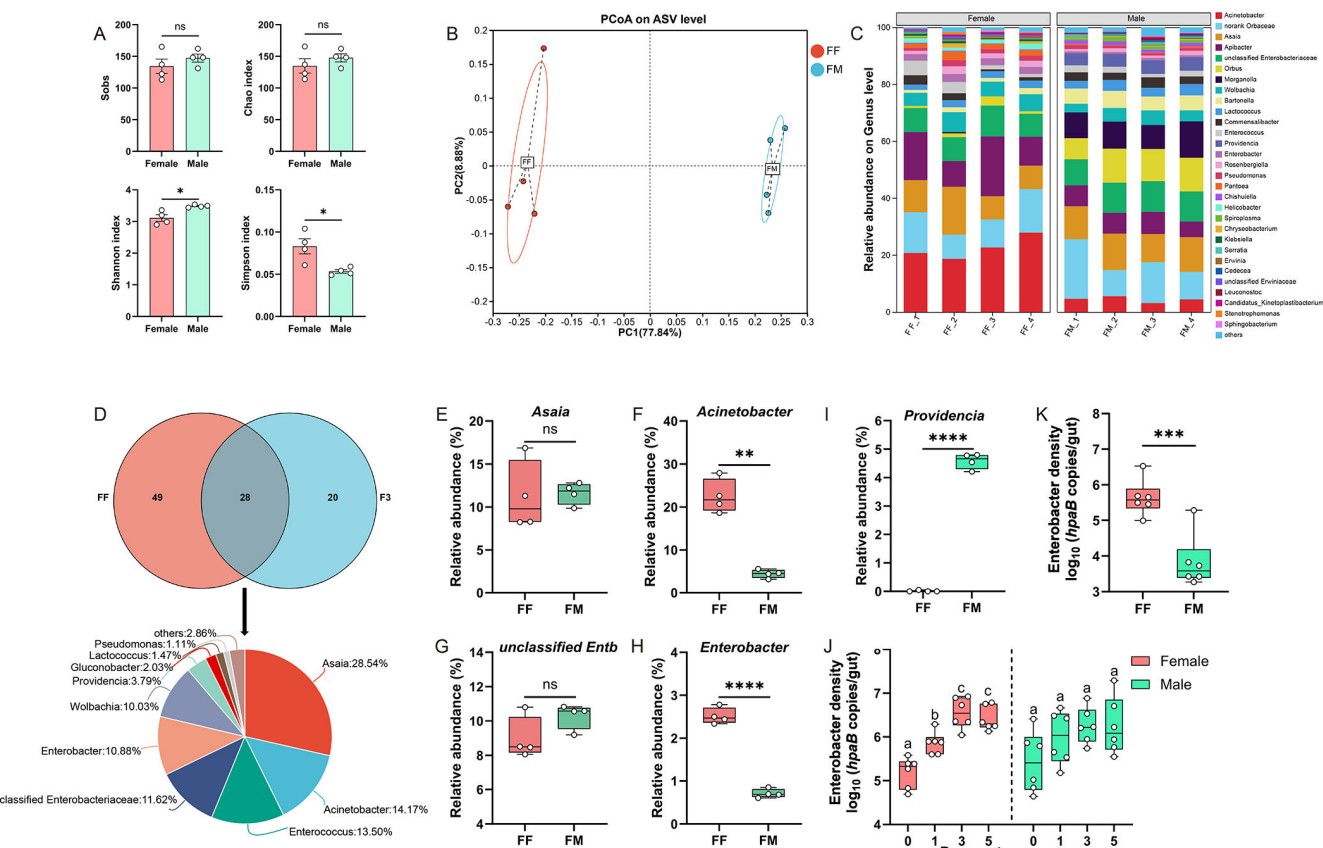

**FIG 3** Gut bacterial community diversity in field *C. medinalis* adults and comparative analysis with laboratory female. (A) Alpha-diversity indices of the gut bacterial community in *C. medinalis* populations of female and male adults collected from the field (*n* = 4). (B) Principal coordinate analysis (PCoA) based on Bray-Curtis dissimilarity. Bacterial community differences between sexes were assessed by PERMANOVA (Adonis test with 999 permutations). Filled circles represent individual samples and represent females (red) and males (green). (C) Relative abundance of bacterial communities at genus levels. Each bar represents one biological replicate (*n* = 4). (D) Venn diagrams of shared genera between field female and laboratory female and their respective proportions. (E–I) Differences in the relative abundance of major shared genera between field female and male, including (E) *Asaia*, (F) *Acinetobacter*, (G) unclassified Enterobacteriaceae, (H) *Enterobacter*, and (I) *Providencia*. (J and K) The absolute abundance of *Enterobacter* in laboratory (J) and field (K) *C. medinalis* adults, respectively. Asterisks indicate statistically significant differences as determined by Student's *t*-test. *$P < 0.05$, **$P < 0.01$, ***$P < 0.001$, ****$P < 0.0001$. Multiple comparisons in panel J were performed using one-way analysis of variance followed by Tukey's post hoc test; different letters indicate significance at $P < 0.05$. Abbreviations: F3 represents females at 3 days post-emergence; FF and FM5 represent field females and field males, respectively; ns, not significant.

of *Enterobacter* increased gradually during ovarian development in laboratory females, whereas no such trend was detected in laboratory males (Fig. 3J). Furthermore, field females harbored a higher absolute abundance of *Enterobacter* than field males (Fig. 3K). Together, these results indicate that although the gut bacterial communities differ significantly between laboratory and field populations of adult *C. medinalis*. They share core genera such as *Enterobacter* that likely mediate a reproduction-promoting function in females across both populations.

## Gut bacteria enhance fecundity of *C. medinalis* females during nutrient stress

*C. medinalis* females were treated with antibiotics to investigate the influence of gut bacteria on reproduction. Antibiotic treatment (ABX) reduced gut bacterial density by 94.99%–96.52% ($P < 0.01$) in females fed different concentrations of honey solution (HS) compared with conventionally reared (CONV) females (Fig. 4A), and almost no culturable bacteria were detected in ABX-treated insects (Fig. 4B). In contrast, ABX had no effect on *Wolbachia* density (Fig. S2A and B). Unexpectedly, under high-nutrition conditions (10% HS), ABX-treated females exhibited significant increases in ovariole length (increase of 23.88%, $P < 0.01$) and ovarian size (increase of 33.68%, $P < 0.01$) compared to CONV-reared females (Fig. 4C through E). Both peak and total egg counts were elevated in response to the ABX treatment, although these increases were not statistically significant (Fig. 4F and H). No significant effects on ovarian morphology or fecundity were observed at 5% HS (moderate nutrition) (Fig. 4C through E and H). In contrast, at 2.5% HS (low nutrition), bacterial elimination significantly reduced ovariole length (decrease of 11.67%, $P = 0.03$) and ovarian size (decrease of 26.47%, $P < 0.01$) (Fig. 4C through E). In addition, bacterial elimination prolonged the preoviposition period (3.48 ± 0.15 to 4.22 ± 0.23 d, $P < 0.01$; Fig. S2C) and decreased both daily fecundity (Fig. 4G) and total fecundity (34.13% reduction, $P < 0.01$; Fig. 4H). However, it did not affect offspring hatching rate or larval feeding consumption (Fig. S2D and E). Because changes in food intake directly affect nutrient availability and may confound reproductive phenotypes, and given that gut microbiota have been reported in regulating insect feeding behavior (24), the feeding amount of *C. medinalis* adults following ABX treatment was further measured. There was no significant change in feeding after gut bacteria were removed in both females and males fed either 10% HS ($P = 0.13$ and 0.50, respectively) or 2.5% HS ($P = 0.76$ and 0.11, respectively) (Fig. S2F through I). Taken together, these findings demonstrated that gut bacteria play a context-dependent role in regulating *C. medinalis* reproduction and can either promote or suppress fecundity, depending on the host nutritional status.

## *Enterobacter bugandensis* from a laboratory population of *C. medinalis* enhances female fecundity

To further investigate the role of gut bacteria in regulating *C. medinalis* reproduction, culturable gut bacteria were isolated from a laboratory population of *C. medinalis* adults and reinoculated them into the host. A total of 234 bacterial strains were identified, and these belonged to 10 genera and 17 species based on BLASTn sequence similarity (Fig. 5A). The most frequently isolated species included *Serratia marcescens*, *Klebsiella michiganensis*, *E. bugandensis*, *Enterobacter cancerogenus*, *Enterococcus casseliflavus*, and *Providencia rettgeri*. In females fed 10% HS and monoassociated with different bacterial strains, monoassociation with *E. bugandensis* significantly increased peak egg production (1.39-fold, $P < 0.01$) relative to the conventionally reared (CONV) group, whereas monoassociation with *P. rettgeri* markedly reduced it (43.88% decrease, $P < 0.01$; Fig. S3A); monoassociation with other bacterial strains had no such effect (Fig. S3A). Both *E. bugandensis* and *P. rettgeri* were able to stably colonize the host gut (Fig. 5B and C). Furthermore, under the 10% HS diet, monoassociation with *E. bugandensis* did not significantly alter total fecundity compared to the CONV group, while *P. rettgeri* still markedly reduced the total number of eggs laid by 51.17% ($P < 0.05$, Fig. 5D). The relative abundance of *P. rettgeri* increased with age in *C. medinalis* females (Fig. S3B and C), which

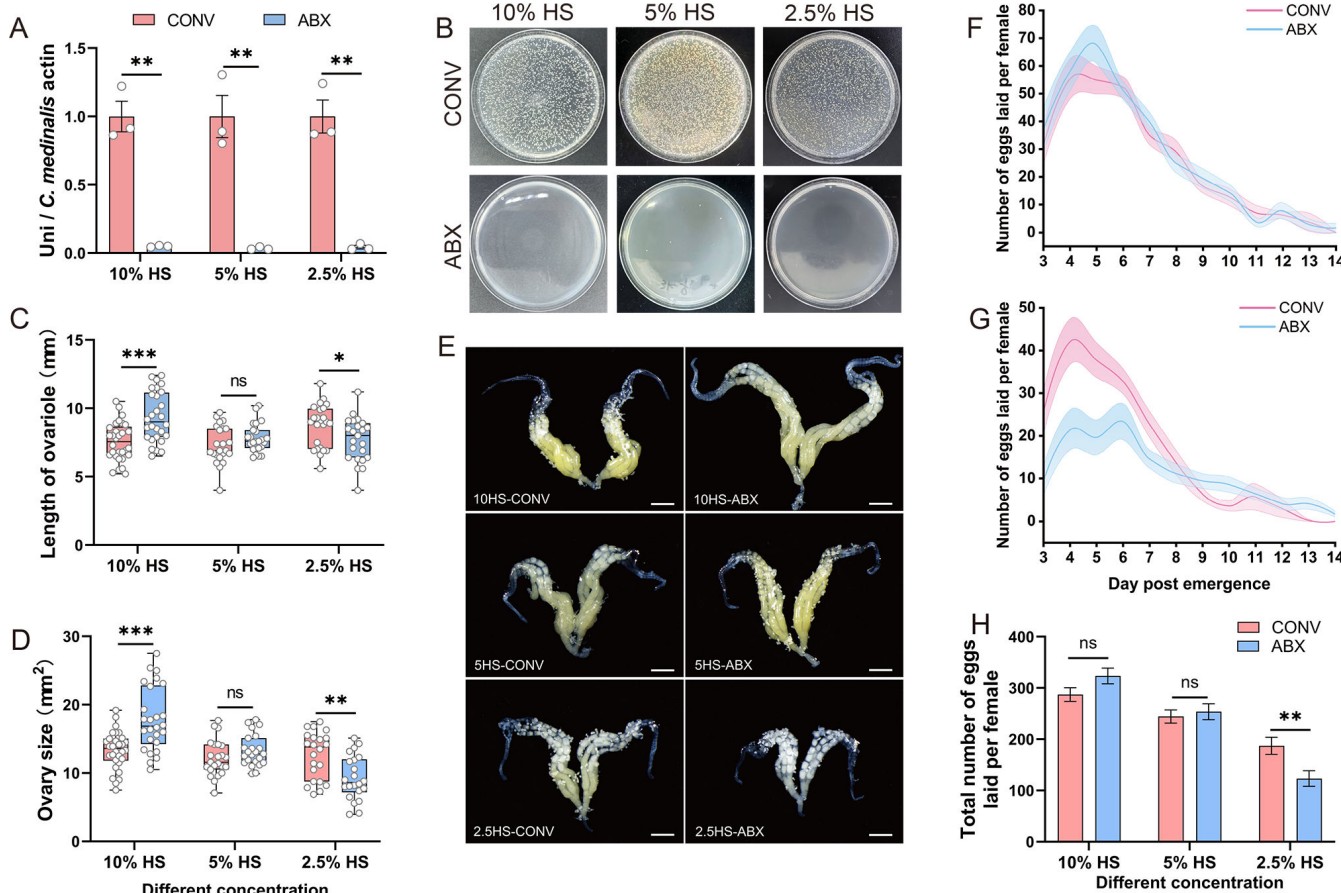

**FIG 4** Elimination of gut microbiota impacts *C. medinalis* reproduction in response to different nutritional conditions. (A and B) Validation of gut bacterial elimination due to antibiotic treatment of *C. medinalis*. (A) qPCR analysis of total bacterial abundance using universal 16S rRNA primers (*n* = 4); values were normalized to *Actin* expression. (B) Inoculation of LB agar with whole-gut homogenates (five guts/group). Upper row shows LB agar inoculated with homogenates from guts of CONV-reared *C. medinalis*; bottom row shows LB inoculated with homogenates from guts of ABX-treated insects. (C–E) Impact of gut microbiota removal on (C) ovariole length, (D) ovarian size, and (E) morphology (scale bars = 1 mm) in CONV- and ABX-treated females reared on HS at 10%, 5%, and 2.5%. (F and G) Impact of gut microbiota presence (CONV, red) and absence (ABX, blue) on the number of eggs laid by females fed with (F) 10% HS and (G) 2.5% HS over a 14-day period. (H) Total number of eggs laid per females supplied with 10%, 5%, and 2.5% HS over a 14-day period after emergence. Asterisks indicate significant differences. *P < 0.05, **P < 0.01, ***P < 0.001. Abbreviations: ABX, antibiotic treatment; CONV, conventionally reared; HS, honey solution; ns, not significant.

may underlie its negative impact on fecundity; however, this strain did not affect the lifespan of *C. medinalis* females (Fig. 5E). When 2.5% HS was used as a nutrient source, supplementation with *E. bugandensis*, which also stably colonized the *C. medinalis* gut under 2.5% HS conditions (Fig. 5F), significantly increased the total number of eggs laid by 1.73-fold (*P* < 0.05), as compared to the ABX group, thus restoring fecundity to a level comparable to the CONV group (Fig. 5G). These results indicate that *E. bugandensis* promotes reproduction in *C. medinalis* when nutrients are limiting.

## *Enterobacter roggenkampii* isolated from field populations of *C. medinalis* enhances female fecundity

To further identify gut bacteria influencing reproduction in *C. medinalis* field populations, bacterial strains were isolated and cultured from adult gut samples, then reinoculated to the insect host. A total of 45 bacterial strains were obtained, and these represented 12 genera and 14 species based on 16S rRNA sequencing. The most abundant species included *S. marcescens*, *Enterococcus faecalis*, *Kluyvera intermedia*, and *Lactococcus lactis* (Fig. 5H). In recolonization experiments with 2.5% HS, monoassociation with *E.*

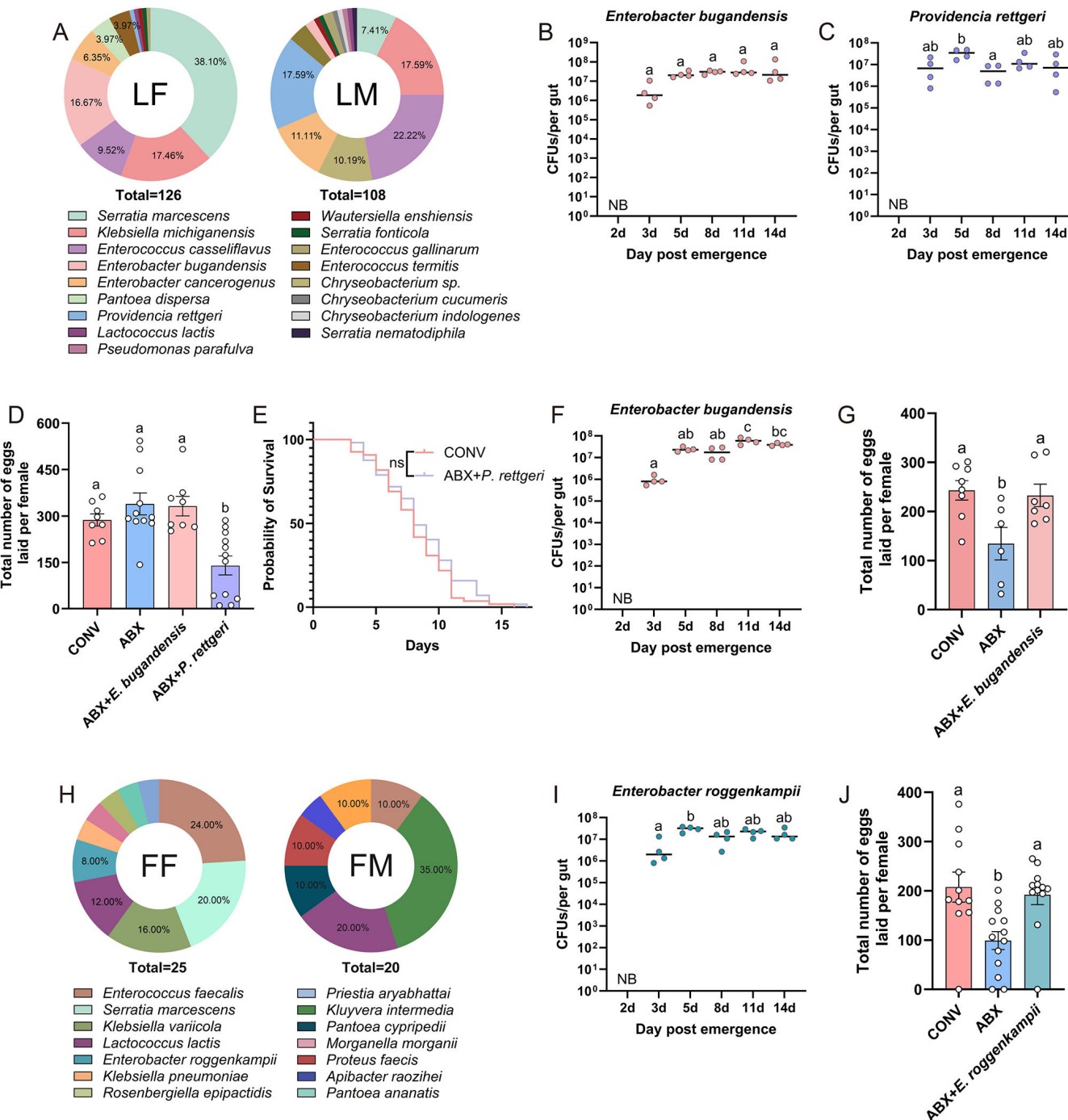

**FIG 5** Species composition of gut symbionts and their contribution to fecundity in *C. medinalis* females. (A) Species structure of culturable bacteria in laboratory female and male adults of *C. medinalis*. (B and C) Colonization of *E. bugandensis* and *Providencia rettgeri* in the gut of females fed with 10% HS. (D) Effect of *E. bugandensis* and *P. rettgeri* supplementation on the total number of eggs laid by females fed with 10% HS. (E) Effect of *P. rettgeri* supplementation on the lifespan of females fed with 10% HS. (F and I) Colonization of *E. bugandensis* and *E. roggenkampii* in the gut of females fed with 2.5% HS. (G) Egg production by CONV-, ABX-, and ABX-treated *C. medinalis* supplemented with *E. bugandensis* and supplied with 2.5% HS. (H) The composition of culturable bacteria in female and male adults from the field. (J) Egg production in CONV-, ABX-, and ABX-treated *C. medinalis* supplemented with *E. roggenkampii* and supplied with 2.5% HS. Multiple comparisons were performed using one-way analysis of variance followed by Tukey's post hoc test; different letters indicate significance at $P < 0.05$. Abbreviations: NB, no bacteria detected; ns, not significant.

*roggenkampii*, which stably colonized the *C. medinalis* gut (Fig. 5I), led to a 1.94-fold ($P <$ 0.05) increase in total egg laying compared to the ABX group, achieving fecundity

equivalent to that of the CONV group (Fig. 5J). In contrast, monoassociation with other commensal bacteria had no significant effect on the number of eggs laid (Fig. S3D). These results indicate that *E. roggenkampii* promotes reproduction in *C. medinalis* when nutrients are limited.

## DISCUSSION

In this study, the diversity and composition of gut bacterial communities were studied in laboratory and field populations of *C. medinalis*. It was found that bacterial richness and diversity indices were higher in field adults than those raised in the laboratory, a phenomenon reported in other insects, including *Drosophila rubida*, *Aedes aegypti*, and *S. frugiperda* (25–27). Taxonomic analysis revealed that the gut bacterial communities in both field and laboratory *C. medinalis* adults were dominated by Proteobacteria and Firmicutes, which agrees with findings in other insects (28). However, field populations of *C. medinalis* adults harbored a more diverse gut microbiome consisting of *Asaia*, *Acinetobacter*, *Apibacter*, unidentified Orbaceae, and unclassified Enterobacteriaceae (Fig. 3). Laboratory populations of *C. medinalis* adults predominantly consisted of *Enterococcus*, *Asaia*, unclassified Enterobacteriaceae, and *Enterobacter*, which collectively accounted for 55%–85% of the microbiota (Fig. 2). The higher microbial diversity in field populations of *C. medinalis* adults is plausibly associated with complex food sources and environmental stresses, and the enriched unique bacterial genera, similar to reports in other insects (29), may enhance the host's adaptability to variable environments.

Gut bacteria associated with reproduction were conserved between field and laboratory populations of *C. medinalis*. In this study, comparative analysis identified several shared core genera, including *Asaia*, *Enterococcus*, *Acinetobacter*, *Enterobacter*, and unclassified Enterobacteriaceae, between field and laboratory females (Fig. 3), suggesting that they represent stable components of the female gut microbiota. Similar conservation of core bacterial taxa between field and laboratory populations has been reported in *Periplaneta americana*, *Blattella germanica*, *Teleogryllus oceanicus*, and *B. dorsalis* (30–33), indicating that maintenance of core microbiota may be a common feature of insect-microbe associations. Core gut symbionts are often involved in host digestion, nutrient assimilation, detoxification, and behavioral regulation (4, 34, 35). In the present study, most shared genera, including *Enterobacter*, were positively correlated with ovarian development or female fecundity. However, the mechanistic basis by which these core gut bacteria influence reproduction in *C. medinalis* remains to be elucidated.

Gut bacteria can enhance fecundity in insects, and this enhancement is critically dependent on the host nutritional status. Previous studies on *D. melanogaster*, *B. dorsalis*, and *Speyeria mormoniae* reported that gut bacteria enhanced fecundity under poor dietary conditions but did not impact reproduction when diets were nutrient rich (11, 13, 36). In this study, it was found that gut bacteria of *C. medinalis* promoted ovarian development and increased reproductive output under low-nutrition conditions (2.5% HS), while no significant effect on reproductive output was observed when nutrition was more plentiful (5% or 10% HS) (Fig. 4). Intriguingly, under high-nutrient conditions (10% HS), antibiotic-treated females exhibited significantly larger ovaries than females with an intact microbiome. Although gut symbionts are known to regulate host feeding (24), the absence of gut bacteria did not alter food intake of *C. medinalis* in this study (Fig. S2). The extra resources for ovarian enlargement may derive from long-term fitness-related traits such as reproduction and longevity, given that a reproduction-longevity trade-off in insects has been well documented in numerous studies (37–39). Therefore, we speculate that antibiotic-induced compensatory development and reduced fecundity are not driven by altered feeding but by microbiota-mediated changes in host resource allocation.

The functions of gut symbionts vary across populations and bacterial strains. Through the identification of culturable bacteria, this study further confirmed that *Enterobacter* was ubiquitous in both laboratory and field *C. medinalis* females, a finding consistent with the results of high-throughput sequencing and qPCR. Among them, *E. bugandensis*

(laboratory isolate) and *E. roggenkampii* (field isolate) both exhibited higher abundance in females than in males, and monocolonization with either strain markedly enhanced *C. medinalis* fecundity under low-nutrient stress (Fig. 5). Combined with previous studies reporting that *E. hormaechei* can boost fecundity in *B. dorsalis* and *S. frugiperda* (12, 13, 40), these findings suggest that the *Enterobacter*-mediated pro-reproductive effect may be conserved across some insect species. Regarding the underlying molecular regulatory mechanisms, gut bacterial symbionts can regulate insects' reproduction through multiple pathways, including metabolite production, mitochondrial activation, and hormonal regulation (3, 40–42), particularly by facilitating the synthesis of 20-hydroxyecdysone (12). However, the specific molecular mechanism by which *E. bugandensis* and *E. roggenkampii* regulate host reproduction remains to be further clarified. Interestingly, in contrast to the reproductive promotion mediated by *Enterobacter*, this study found that the commensal bacterium *P. rettgeri* decreased fecundity of *C. medinalis* (Fig. 5). While lethal effects of *P. rettgeri* have been reported in several insects (43–45), no lethal impact on females was observed in this study, indicating that its fecundity-inhibiting effect arises from a unique molecular mechanism rather than toxicity. Collectively, these findings show that the reproductive effects of gut symbionts in *C. medinalis* depend on bacterial strain and host population context, underscoring microbial and ecological variability in host-microbe interactions.

In summary, these results demonstrate that distinct gut bacterial communities exist in laboratory and field populations of *C. medinalis* adults, with shared core *Enterobacter* taxa strongly linked to female reproduction. Furthermore, under nutrient-poor conditions, gut symbionts enhance host reproduction in *C. medinalis*, and *E. bugandensis* and *E. roggenkampii* are important in conferring the increased fecundity. The underlying mechanisms used by gut bacteria to modulate *C. medinalis* reproduction require further inquiry. It is plausible that these findings may ultimately reveal a novel pest control strategy where populations of essential gut bacteria are targeted by antimicrobial agents, e.g., narrow-spectrum antibiotics, bacteriophages, or bacteriocins, that ultimately suppress pest populations.

## MATERIALS AND METHODS

### Rearing of insects

The larvae of leaf folders were obtained from experimental rice fields at Yangzhou University (Jiangsu, China) and were maintained on seedlings of wheat until pupation. After eclosion, male/female pairs were moved to transparent cups (500 mL). Cotton wool saturated in a 10% HS served as nutritional supplement. Plastic film was used to cover the cups and served as a surface for oviposition. Insects were maintained in climate-controlled chambers at 27°C ± 1°C, 70% ± 5% RH and a 14 h light/10 h dark photoperiod.

### Gut dissection and bacterial DNA extraction

Prior to dissection, the wings of moths were removed with scissors, and surface scales were gently brushed away with a soft brush. The moths were then surface-sterilized for 3 min in 75% ethanol and rinsed in PBS (pH 7.4). The entire gut starting at the crop and extending to the hindgut (minus Malpighian tubules) was dissected in ice-cold PBS using sterile tools. The dissected gut tissues were subsequently used for DNA extraction.

### Sequencing and analysis of bacterial 16S rDNA

DNA was extracted from guts (*n* = 30 per biological replicate, four replicates in total) using a Stool DNA Extraction Kit (SimGEN, Hangzhou, China) as recommended by the manufacturer. DNA purity and quantity were determined with a spectrophotometer (NanoDrop 2000 UV-vis; NanoDrop Technologies, Wilmington, DE, USA), and the integrity

of DNA was assessed by agarose (1%) gel electrophoresis. The V3–V4 hypervariable segments in the 16S rRNA of bacteria were amplified with primers 338F and 806R (Table S1). Triplicate samples of the amplified products were combined and purified with the AxyPrep DNA Gel Extraction Kit (Corning Incorporated, New York, USA). Library construction utilized the NEXTFLEX Rapid DNA-Seq Kit (Revvity, Boston, MA, USA), and the Illumina MiSeq platform in Majorbio Bio-Pharm Technology Co., Ltd. (Shanghai, China) was used for sequencing. Raw sequence data were deposited at the National Center for Biotechnology Information (NCBI) as accession numbers PRJNA1302693 and PRJNA1416006.

The same data processing and analysis procedures were applied to the sequencing data of laboratory populations and the field population of *C. medinalis*. Raw and paired-end sequences were filtered and assembled with fastp (v0.20.0) and FLASH (v1.2.7), respectively. Based on the default parameters, the DADA2 module within QIIME2 (v2022.02) was employed for denoising and yielding preliminary amplicon sequence variants (ASVs). These ASVs were taxonomically classified using RDP Classifier (v2.2) against the SILVA 138 16S bacterial database, with a confidence threshold of 0.7. Subsequently, sequences annotated as mitochondria or chloroplast DNA were eliminated from the data set prior to further analysis. Analysis of data occurred on the Majorbio Cloud platforms. Alpha diversity was analyzed using mothur. PCoA was used to evaluate beta diversity using weighted UniFrac distances. The statistical significance of differences between groups was assessed via PERMANOVA. LEfSe analysis was executed to detect taxa with significantly different abundances (linear discriminant analysis score >2, $P <$ 0.05) from phylum to genus levels among groups. Heatmap analysis of species abundance and reproductive indices was performed using Spearman correlations ($|r| > 0.6$, $P <$ 0.05).

## Culturing, quantifying, and identifying gut bacteria

Gut samples were dissected from 30 female and male moths sampled from laboratory (3 days post-emergence) and field populations of *C. medinalis*. Samples were emulsified in 1 mL of sterile PBS with a tissue homogenizer (Shanghai Jingxin Industrial Development Co., Ltd., Shanghai, China) at 70 Hz/s for 60 s. Serial dilutions of the homogenate were inoculated to Luria-Bertani (LB) agar and incubated for 24–48 h at 30℃. Individual colonies were isolated and inoculated into LB broth with an overnight incubation at 30℃ and agitation at 220 rpm. Bacterial DNA Extraction Kit (SimGEN) was used to extract DNA, and genes encoding 16S rRNA were amplified by PCR with universal primers 27F and 1492R (Table S1). PCR was conducted as follows: initial denaturation, 94℃ for 3 min; 35 cycles at 94℃ for 10 s; annealing at 55℃, 10 s; extension at 72℃, 30 s; and final extension at 72℃, 5 min. PCR products were purified with the E.Z.N.A. Cycle-Pure Kit (Omega, Norcross, GA, USA) and sequenced using the bidirectional Sanger approach. Sequences were then subjected to BLAST analysis using the NCBI 16S rRNA database.

## Quantification of bacterial density by qPCR

Bacterial DNA was isolated from the guts of adult moths as described above. Quantitative PCR was performed to determine the total bacterial density and *Enterobacter* genus density in adult guts using Uni331F/Uni797R and hpa287F/hpa625R primers, respectively (primer sequences see Table S1). Two distinct plasmids were constructed and cloned, each harboring the bacterial universal 16S rDNA fragment and the *Enterobacter*-specific *hpaB* gene fragment, respectively. These plasmids were used to generate standard curves for absolute quantification of gene copy numbers. For samples from the antibiotic treatment experiment, the expression level of total bacterial 16S rDNA was determined by relative quantification and normalized using the *C. medinalis* Actin gene (primers are listed in Table S1). Each 10 µL qPCR reaction contained the following: SYBR Green Mix, 5 µL (Vazyme, Nanjing, China); primers, 100 nM each; and DNA template, 5 ng. Reactions were run on a Bio-Rad CFX Connect system (Bio-Rad, Hercules, CA, USA)

as follows: preincubation at 50°C for 2 min, followed by 95°C for 10 min; denaturation at 95°C for 15 s and 45 cycles; annealing/extension for 1 min at 60°C; and melting curve analysis, at 95°C for 15 s, 53°C for 15 s, and 95°C for 15 s.

## Antibiotic treatments and phenotypes of *C. medinalis* ovaries

Newly emerged *C. medinalis* moths were supplied with HS (10%, 5%, or 2.5%) on cotton pads that were replaced every 24 h and amended with penicillin, gentamicin, and streptomycin at 1.5, 1.5, and 2.5 mg/mL, respectively; the same antibiotic concentrations were used throughout this study. Adult female *C. medinalis* were anesthetized with 75% ethyl alcohol and dissected in cold PBS, and ovarian phenotypes were assessed for different developmental stages and treatments. Images of the ovaries were captured using a stereomicroscope equipped with a digital camera (Olympus, model SZX23, Japan), and measurements of ovarian size and ovariole length were conducted with mShot software (Mingdao, Guangzhou, China). Ovarian development in *C. medinalis* was assigned to one of the following five maturation stages: transparent opalescent phase, stage I; yolk deposition, stage II; egg maturation, stage III; peak oviposition phase, stage IV, and a terminal phase, stage V. The proportions of different stages were determined as the quantity of ovaries at a given stage divided by the total number of ovaries examined.

## Assessment of moth fecundity, gut bacterial persistence, and longevity

Newly emerged moths were either treated with antibiotics (ABX group) or left untreated (conventionally treated, CONV group) for 2 days. The moths were provided with culturable bacteria via cotton balls soaked in 2.5% or 10% HS containing bacterial suspensions at an $OD_{600} = 2.0$ for 3 days or with sterile HS as a control.

For fecundity (quantity of eggs laid) testing, a single female-male pair was then transferred to a self-designed egg-counting device. The egg numbers deposited in each container were recorded daily until 14 days post-emergence, and total egg counts were calculated. All experiments included three or more biological replicates.

To determine gut bacterial persistence, female moths were dissected on day 2 post-antibiotic treatment and on days 1, 3, 6, 9, and 12 post-bacterial feeding, respectively. Dissected gut tissues ($n = 4$ guts per biological replicate, four replicates per time point) were homogenized in 300 µL PBS for 60 s using a tissue homogenizer. Samples were serially diluted, and dilutions were then plated (2.5 µL) onto a CHROMagar Orientation chromogenic medium, and CFUs were counted after 1 day of cultivation.

For longevity assessment, moths fed with *P. rettgeri* or sterile 10% HS (CONV group) were monitored daily at a fixed time; dead moths were counted and removed.

## Statistical analyses

GraphPad Prism (v10.0) and SPSS 31.0 were used to calculate statistical differences. Comparisons between groups were evaluated with two-tailed Student's *t*-tests. When multiple groups were compared, one-way analysis of variance was performed followed by Tukey's post hoc test. Data points were determined as statistically significant when $P < 0.05$. Survival curves were analyzed via the Kaplan-Meier method, with intergroup significance evaluated by the log-rank test. Data points are shown as means ± SEM, and further information on statistical analyses is provided in the figure legends.

## ACKNOWLEDGMENTS

This research was financially supported by the Top Talents Award Plan of Yangzhou University (2023) and the National Natural Science Foundation of China (no. 32302436).

F.L., Q.P., and Z.Y. conceived and designed the project. Z.Y., C.X., and Y.A. performed the experiments and analyzed data. Y.A., Y.L., Z.W., and X.L. executed the experiments with antibiotics and counted the number of eggs laid by *C. medinalis*. Z.Y., Q.P., and F.L. wrote the manuscript. All authors read and approved the final manuscript.

## AUTHOR AFFILIATIONS

[1]College of Plant Protection, Yangzhou University, Yangzhou, Jiangsu, China

[2]Joint International Research Laboratory of Agriculture and Agri-Product Safety, Ministry of Education of China, Yangzhou University, Yangzhou, Jiangsu, China

[3]Institute of Agricultural Science and Technology Development, Yangzhou University, Yangzhou, Jiangsu, China

## AUTHOR ORCIDs

Zhichao Yao  http://orcid.org/0000-0001-5537-7102
Fang Liu  http://orcid.org/0000-0002-1907-6019
Qinjian Pan  http://orcid.org/0000-0001-9045-672X

## DATA AVAILABILITY

Sequencing data have been deposited in the NCBI SRA database (PRJNA1302693 and PRJNA1416006).

## ADDITIONAL FILES

The following material is available online.

### Supplemental Material

**Supplementary material (Spectrum03878-25-s0001.docx).** Supplemental methods; Fig. S1 to S3; Table S1.

### Open Peer Review

**PEER REVIEW HISTORY (review-history.pdf).** An accounting of the reviewer comments and feedback.

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
