## [Reviewer comments · Microbiology Spectrum]

Microbiology Spectrum

Diversity of the *Cnaphalocrocis medinalis* gut bacterial community and its contribution to reproduction

Zhichao Yao, Chengrui Xue, Yang Ang, Ziwei Wu, Xin Liu, Ying Liu, Fang Liu, and Qinjian Pan

Corresponding Author(s): Qinjian Pan, Yangzhou University

Review Timeline:

Submission Date:	December 2, 2025
Editorial Decision:	December 22, 2025
Revision Received:	February 18, 2026
Accepted:	March 12, 2026

Editor: John Chaston

Reviewer(s): Disclosure of reviewer identity is with reference to reviewer comments included in decision letter(s). The following individuals involved in review of your submission have agreed to reveal their identity: Letian Xu (Reviewer #2)

Transaction Report:

DOI: <https://doi.org/10.1128/spectrum.03878-25>

Re: Spectrum03878-25 (**Diversity of the *Cnaphalocrocis medinalis* gut bacterial community and its contribution to reproduction**)

Dear Dr. Qinjian Pan:

Thank you for the privilege of reviewing your work. Below you will find my comments, instructions from the Spectrum editorial office, and the reviewer comments.

The reviewers are both generally positive about the work and its quality, but also identified some areas of concern. As you address these concerns, please consider their comments carefully, as I think they will be useful to you. Also, as you address these concerns please remember that textual modification of interpretations may be an appropriate substitution for new experiments. For example, reviewer 1 suggests in points 3 and 9 that new experiments or analyses that could be included. These may be very useful, but an alternative may be to adjust the conclusions and interpretations. Please consider appropriately, and I hope you choose to submit a revised version of the manuscript.

Revision Guidelines

Sincerely,
John Chaston
Editor
Microbiology Spectrum

Reviewer #1 (Comments for the Author):

The paper focuses on the study of gut microbiota in adult rice leaf roller *Cnaphalocrocis medinalis*, providing certain significance and value in analyzing the composition and potential functions of their gut microbial community. The reported potential pathogenic microorganism *Providencia rettgeri* that inhibit the reproduction of rice leaf roller may serve as a reference for biological control or microbial management. However, there are numerous issues in the research methods, data analysis, results presentation, and discussion, indicating that the overall quality needs improvement.

A major revision or consideration for rejection is recommended. Below are specific comments:

1. **Wolbachia-related:** In the bar chart of Figure 2C, a significant presence of the endosymbiont *Wolbachia* (indicated by purple icons) is observed in all samples, with an abundance of approximately 20%. *Wolbachia* is a famous endosymbiont that is prevalent and manipulates host reproduction in many arthropods. Since the localization of *Wolbachia* and other gut bacteria may differ within the insect's body, it is highly likely that the antibiotic treatment also affected *Wolbachia*. It is better to monitor this aspect and provide necessary explanations and analyses in the text to clarify the potential impact of this factor on the experimental results.

2. **Field Sample Microbial Community Analysis:** The analysis of the gut microbial community in field samples shows that some microorganisms frequently appear, which is an important finding indicating that there may be a shared core microbiota between laboratory and wild populations. It is recommended that the authors conduct an in-depth discussion on this significant finding and further optimize the research design, possibly focusing on these microorganisms as key research subjects to explore their roles in the physiology of rice leaf roller moths.

3. **Analysis of Shared Microbial Communities:** To clarify the shared microbial communities between laboratory and wild populations, the authors may consider creating a Venn diagram to merge and analyze the microbial communities obtained from the two populations. This approach would provide a clearer and more intuitive presentation of the similarities and differences in microbial communities between the two populations, offering stronger data support for subsequent research, for example, justifying studying on *Enterobacter bugandensis* and *Enterobacter roggenskampii*. Gut microbes of lepidoptera are easily changed by environments. therefore, the significance of studying certain gut bacteria is limited.

4. **Acquisition Method of Experimental Populations:** The method of acquiring populations in the experiments related to Figure 4 deserves attention. According to the materials and methods, the authors conducted mating and oviposition experiments two days after feeding newly emerged adults with antibiotics, observing the offspring's oviposition rate. This method presents the overall effect of antibiotic treatment and cannot be solely attributed to changes in microbial communities, as antibiotic treatment directly affects the insects themselves. Typically, such experiments require a recovery period of at least 2-3 generations to minimize the impact of antibiotic treatment on the insects, thus allowing for a more accurate assessment of the role of microbial communities in insect biology. It is suggested that the authors adjust the experimental design, conduct relevant experiments again, or adequately discuss this limitation in the text.

5. **Noise Reduction and ASV Generation Methods:** In line 299 of the paper, the authors should specify the exact methods used for noise reduction and ASV generation in QIIME2. The text mentions that it is neither Dada2 nor Deblur but does not clarify the actual method used, which could affect the reproducibility and scientific validity of the research. The authors must clearly articulate the methods employed.

6. **Differences in Sequencing Companies and Methods:** The laboratory samples in this study were sequenced by Shanghai Majorbio Bio-Pharm Technology Co., Ltd., while the field samples were sequenced by Tianjin Novogene Co., Ltd. The sequencing platforms and library preparation techniques of the two companies may differ, raising questions about the comparability of the data and affecting the reliability of the research results. The authors are advised to clarify whether they used their own methods for data analysis; if they used the analysis processes provided by the companies, they should detail them separately to ensure the reproducibility of the data analysis results and clearly compare the differences between the two. Additionally, the authors did not mention this significant issue of different sequencing methods for the two batches of samples in the discussion section, while sequencing batch effects are crucial in microbiome analysis. Given the current analysis methods, the comparability of results between the two batches is questionable. The authors should carefully review the analysis process to avoid drawing incorrect conclusions. If possible, the field sample data could also be considered for removal, as it contributes little to the overall paper.

7. **PCA Plot and Statistical Analysis:** The PCA plot in Figure 2 is recommended to be redrawn. Since the samples involve two dimensions, i.e. gender and adult age, to clarify which factor has a greater impact on the microbial community composition, it is suggested to use different colors or shapes to denote these two factors. Additionally, the authors' PERMANOVA test should adopt a similar two-factor analysis method to show the contribution of gender or age factors to the microbial community composition. Furthermore, the statistical analysis methods in Figure 2A should ideally be explained in the figure legend to enhance readers' understanding of the data.

8. **Primer Citation:** In Table S1, the primers 338F and 806R do not first appear in Liu's 2016 paper; the authors should cite the

original literature. Additionally, the primers 27F and 1492 are classic and widely used universal bacterial primers, not first proposed in Yao's 2022 paper. The authors should provide appropriate references for accurate citation to ensure the rigor and scientific validity of the research.

9. qPCR Analysis: The primers provided in this paper, except for actin, are not suitable for qPCR analysis of bacterial abundance. Given that the authors used qPCR technology, it is recommended to conduct absolute quantification PCR for key bacteria (such as *Enterobacter bugandensis*, *Enterobacter roggkampii*, and *Providencia rettgeri*) with significant changes in abundance. This approach will help determine whether there are significant differences between males and females or between laboratory and field samples. Compared to the relative abundance obtained from 16S rRNA amplicon sequencing, absolute quantification PCR can more accurately reflect the actual quantity of bacteria, avoiding the influence of proportional differences in sample data on the results.

10. Antibiotic Treatment Experimental Results: In the antibiotic treatment experiments, the authors fed newly emerged rice leaf roller moths with honey water containing three different concentrations of antibiotics, but the paper does not present the differences in results among the three antibiotic treatments nor specify which population was used after antibiotic treatment. This makes the presentation of experimental results incomplete and fails to clearly demonstrate the impact of antibiotic concentration on the results. It is suggested that the authors supplement relevant results, clarify the differences between treatment groups, and specify the populations used to help readers better understand the experimental conclusions.

11. Ecological Experimental Results: Some ecological experimental results in the supplementary table, such as the analysis of the feeding content of rice leaf roller moths, are interesting, but the authors did not indicate or discuss the reasons for conducting this experiment and its special biological significance, making it difficult for readers to understand the relevance of this experiment to the overall study. It is recommended that the authors provide explanations and discussions on these ecological experimental results in appropriate sections of the text to enhance the logical coherence of the research.

12. Figure 5 Legend Presentation: In Figure 5, Figures 5C and 5D analyze the oviposition quantity and other indicators of insects after bacterial infection, and this experiment is a one-way ANOVA. The current legend presentation is not easy to understand and may cause confusion. It is suggested that the authors use common labeling methods such as A, B, C, D to represent different treatment groups, making the legend clearer and easier for readers to interpret the experimental results.

13. In lines 251 and 252, the authors suggest that nutritional status and gut microbiota synergistically activate the nutritional and hormonal signaling pathways in rice leaf roller moths, but lack supporting evidence for this claim, indicating an overinterpretation in the discussion. Moreover, the entire discussion section lacks clarity in logic, and the causal relationships between the evidence chains are not direct, making it difficult for readers to understand the connection between the research results and conclusions. It is recommended that the authors reorganize the discussion section content to present research findings and their significance with more rigorous logic and sufficient evidence, avoiding overinterpretation and vague causal relationship statements.

In summary, it is hoped that the authors can comprehensively and thoroughly address the above issues to improve the research content and paper presentation, thereby enhancing the quality of the paper. If the authors cannot effectively resolve these issues, consideration for rejection may be necessary.

Reviewer #2 (Comments for the Author):

This study investigates the gut bacterial community structure and its relationship with reproduction in lab and field populations of the rice leaf folder, *Cnaphalocrocis medinalis*. The authors found that during ovarian development, bacterial load and the abundance of Enterobacteriaceae significantly increased and positively correlated with fecundity. Using antibiotic treatment (ABX) and monoassociation, the study demonstrated that gut bacteria regulate reproduction in a nutrient-context dependent manner: inhibiting it under low nutrition but showing compensatory ovarian growth under high nutrition. Key species like *Enterobacter bugandensis* and *E. roggkampii* were identified as reproductive promoters. The study addresses an important ecological question, but the manuscript requires improvements in logical consistency, citation accuracy, and mechanistic depth. Specifically, the conflict in the introduction must be resolved, and the discussion on molecular pathways (like 20E synthesis) needs to be strengthened with recent literature and, ideally, preliminary validation.

Major Comments

1. Logical Conflict in Introduction (Lines 56-58): The statement in lines 56-58 suggesting that gut microbiota have no significant effect on adult lepidopteran biology directly contradicts the following paragraph, which describes how gut bacteria promote reproduction in *Spodoptera frugiperda*. This transition is jarring. I recommend either rephrasing this section or deleting the "no effect" claim in lines 56-58 to maintain a consistent logical flow toward the role of microbes in reproduction.

2. Molecular Mechanisms and 20E Synthesis (Lines 249-250): When discussing hormonal regulation, the authors should consider whether the gut microbiota promotes reproduction by facilitating the synthesis of 20-hydroxyecdysone (20E). I suggest adding this point to lines 249-250 and citing recent evidence (e.g., *Microbiome* (2025) doi.org/10.1186/s40168-025-02278-5).

Furthermore, experimental validation (e.g., qPCR of Vg genes) is still highly recommended.

3. Compensatory Development under High Nutrition: The phenomenon where ABX treatment leads to increased ovary size under 10% HS is intriguing. The authors should discuss whether this represents a fitness trade-off (e.g., at the cost of longevity) rather than just reporting the morphological change.

4. Data on the colonization dynamics of re-introduced bacteria are needed. Additionally, survival curves for the *P. rettgeri* group must be provided to ensure that its inhibitory effect on reproduction is not simply a side effect of host morbidity/pathogenicity.

Minor Comments

1. Line 70): Reference 16 cited in line 70 focuses on insecticide detoxification, which is not directly relevant to the current discussion on host reproduction. Please replace it with more appropriate references, such as doi.org/10.1038/s41522-023-00435-y and [doi: 10.1111/1749-4877.12911](https://doi.org/10.1111/1749-4877.12911).

2. Word Choice (Line 64): Change "poorly understand" to "elusive."

3. Figure Clarity: Improve the visibility of scale bars in Figure 1C and increase font sizes in the Figure 2H heatmap.

4. Feeding Stats: Explicitly report statistical values (e.g., P-values) in the Results to confirm that ABX treatment does not influence food intake.

Reviewer #1 (Comments for the Author):

The paper focuses on the study of gut microbiota in adult rice leaf roller *Cnaphalocrocis medinalis*, providing certain significance and value in analyzing the composition and potential functions of their gut microbial community. The reported potential pathogenic microorganism *Providencia rettgeri* that inhibit the reproduction of rice leaf roller may serve as a reference for biological control or microbial management. However, there are numerous issues in the research methods, data analysis, results presentation, and discussion, indicating that the overall quality needs improvement.

A major revision or consideration for rejection is recommended. Below are specific comments:

1. *Wolbachia*-related: In the bar chart of Figure 2C, a significant presence of the endosymbiont *Wolbachia* (indicated by purple icons) is observed in all samples, with an abundance of approximately 20%. *Wolbachia* is a famous endosymbiont that is prevalent and manipulates host reproduction in many arthropods. Since the localization of *Wolbachia* and other gut bacteria may differ within the insect's body, it is highly likely that the antibiotic treatment also affected *Wolbachia*. It is better to monitor this aspect and provide necessary explanations and analyses in the text to clarify the potential impact of this factor on the experimental results.

Re: We sincerely appreciate the reviewer's insightful comment on the potential impact of antibiotic treatment on *Wolbachia*. *Wolbachia* is important endosymbiont in insects. Firstly, we selected antibiotics penicillin, streptomycin, gentamicin to target extracellular gut bacteria, but not the classic antibiotics (tetracycline, rifampicin) used to eliminate intracellular *Wolbachia*. Secondly, our antibiotic treatment is short-term (4-days), and may have minimal effects on *Wolbachia*. To verify this, we added an qPCR experiment targeting the *wsp* gene and found no significant difference in

Wolbachia copy numbers between treated and control groups (see lines 169 on page 7 of the revised manuscript; Fig. S2A, B in Supplemental Material), confirming that *Wolbachia* load remained unchanged.

2. Field Sample Microbial Community Analysis: The analysis of the gut microbial community in field samples shows that some microorganisms frequently appear, which is an important finding indicating that there may be a shared core microbiota between laboratory and wild populations. It is recommended that the authors conduct an in-depth discussion on this significant finding and further optimize the research design, possibly focusing on these microorganisms as key research subjects to explore their roles in the physiology of rice leaf roller moths.

Re: We greatly appreciate the reviewer's insightful comments and valuable suggestions. Following the reviewer's advice, we conducted a comparative analysis of gut microbial communities between field and laboratory populations of *C. medinalis*. We found that the existence of several shared core bacterial genera across both populations, as the reviewer pointed out (see Fig. 3D; lines 151-153 on page 7 of the revised manuscript). We have added an in-depth discussion of this key finding (see lines 236-247 on page 10 of the revised manuscript) and optimize the research design (see lines 160-163 on page 7 of the revised manuscript) to focus on these shared core microorganisms as key subjects exploring their roles in the physiology of *C. medinalis*.

3. Analysis of Shared Microbial Communities: To clarify the shared microbial communities between laboratory and wild populations, the authors may consider creating a Venn diagram to merge and analyze the microbial communities obtained from the two populations. This approach would provide a clearer and more intuitive presentation of the similarities and differences in microbial communities between the two populations, offering stronger data support for subsequent research, for example, justifying studying on *Enterobacter bugandensis* and *Enterobacter roggkampii*. Gut microbes of lepidoptera are easily changed by environments. therefore, the significance

of studying certain gut bacteria is limited.

Re: Thanks for reviewer's insightful comments. Following the advice, we added a Venn diagram analysis to compare gut microbial communities between laboratory and field populations, identifying several shared bacterial genera in female adults of two populations (see Fig. 3D; lines 151-153 on page 7 of the revised manuscript). Interestingly, we found that *Enterobacter* exhibited distinct sex- and development-specific patterns in both the relative abundance and absolute abundance (See Fig. 2G and Fig. 3H, 3J, 3K; lines 130-132 on page 6 and lines 153-160 on page 7 of the revised manuscript), showing a positive correlation with female reproduction (See Fig. 2I; lines 133-135 on page 6 of the revised manuscript). These findings provide stronger data support and solid justification for our subsequent functional studies on *Enterobacter bugandensis* and *Enterobacter roggenkampii*.

4.Acquisition Method of Experimental Populations: The method of acquiring populations in the experiments related to Figure 4 deserves attention. According to the materials and methods, the authors conducted mating and oviposition experiments two days after feeding newly emerged adults with antibiotics, observing the offspring's oviposition rate. This method presents the overall effect of antibiotic treatment and cannot be solely attributed to changes in microbial communities, as antibiotic treatment directly affects the insects themselves. Typically, such experiments require a recovery period of at least 2-3 generations to minimize the impact of antibiotic treatment on the insects, thus allowing for a more accurate assessment of the role of microbial communities in insect biology. It is suggested that the authors adjust the experimental design, conduct relevant experiments again, or adequately discuss this limitation in the text.

Re: Thanks for this insightful comments. Many research on agricultural pests used method of assessing antibiotic-induced phenotypic changes within a single generation (e.g., Zhang et al., Cell Reports, 2025, 44(7):115911; Zhang et al., ISME J, 2025, 19(1):wraf237; Hammer et al., 2017, PNAS, 114(36):9641-9646). We referred to these methods to design our experiments and to examine regulatory effects of the gut

microbiota on host reproduction.

Antibiotic treatment may directly affect the insects themselves. Our results showed that reintroducing symbiotic bacteria restored egg-laying capacity under low-nutrient conditions (see Fig. 5G, 5J; lines 204-207 and lines 215-218 on page 9 of the revised manuscript), indicating that antibiotic treatment hardly affects *C. medinalis*.

5.Noise Reduction and ASV Generation Methods: In line 299 of the paper, the authors should specify the exact methods used for noise reduction and ASV generation in QIIME2. The text mentions that it is neither Dada2 nor Deblur but does not clarify the actual method used, which could affect the reproducibility and scientific validity of the research. The authors must clearly articulate the methods employed.

Re: Thank you for pointing out our mistake. After carefully reviewing our sequence processing methods, we have confirmed that "DADA2 (v1.14)" was used for denoising and generating Amplicon Sequence Variants (ASVs) accordingly (See lines 315-317 on page 13 of the revised manuscript).

6.Differences in Sequencing Companies and Methods: The laboratory samples in this study were sequenced by Shanghai Majorbio Bio-Pharm Technology Co., Ltd., while the field samples were sequenced by Tianjin Novogene Co., Ltd. The sequencing platforms and library preparation techniques of the two companies may differ, raising questions about the comparability of the data and affecting the reliability of the research results. The authors are advised to clarify whether they used their own methods for data analysis; if they used the analysis processes provided by the companies, they should detail them separately to ensure the reproducibility of the data analysis results and clearly compare the differences between the two. Additionally, the authors did not mention this significant issue of different sequencing methods for the two batches of samples in the discussion section, while sequencing batch effects are crucial in microbiome analysis. Given the current analysis methods, the comparability of results between the two batches is questionable. The authors should carefully

review the analysis process to avoid drawing incorrect conclusions. If possible, the field sample data could also be considered for removal, as it contributes little to the overall paper.

Re: We acknowledge the reviewer's concern on variations in sequencing platforms, library preparation, and analytical pipelines. To address this, we re-sequenced all field samples at Shanghai Majorbio Bio-Pharm Technology Co., Ltd., using the same sequencing platform, library preparation protocol, quality control standards, and downstream bioinformatic pipeline as for the laboratory samples. We have added the Methods section with detailed unified protocols to guarantee data reproducibility and reliability (See lines 313-327 on page 13 of the revised manuscript). We then re-analyzed the merged dataset under a unified framework, which eliminated batch effects and ensured full comparability between the two populations (See Fig. 3D, lines 151-153 on page 7 of the revised manuscript; Fig. S1B in Supplemental Material).

7.PCA Plot and Statistical Analysis: The PCA plot in Figure 2 is recommended to be redrawn. Since the samples involve two dimensions, i.e. gender and adult age, to clarify which factor has a greater impact on the microbial community composition, it is suggested to use different colors or shapes to denote these two factors. Additionally, the authors' PERMANOVA test should adopt a similar two-factor analysis method to show the contribution of gender or age factors to the microbial community composition. Furthermore, the statistical analysis methods in Figure 2A should ideally be explained in the figure legend to enhance readers' understanding of the data.

Re:Thanks. Following reviewer's recommendation, we have redrawn the PCoA plot in Figure 2, using distinct colors and shapes to distinguish age and gender. In addition, we performed a two-factor PERMANOVA, revealing that age is the dominant factor shaping microbial community composition, while gender has negligible influence (See Fig. 2B; lines 118-119 on page 5 of the revised manuscript). Furthermore, we re-analyzed the diversity indices in Figure 2A by conducting separate one-way ANOVAs for male and female groups, with Tukey's post hoc tests; the detailed

methods are now described in the figure legend (See lines 570-572 on page 23 of the revised manuscript).

8. Primer Citation: In Table S1, the primers 338F and 806R do not first appear in Liu's 2016 paper; the authors should cite the original literature. Additionally, the primers 27F and 1492 are classic and widely used universal bacterial primers, not first proposed in Yao's 2022 paper. The authors should provide appropriate references for accurate citation to ensure the rigor and scientific validity of the research.

Re: Thank you for your valuable comment. As suggested, we have now revised the manuscript by providing the appropriate original citations for primers 338F/806R and 27F/1492R in Table S1 (See Supplemental Material).

9. qPCR Analysis: The primers provided in this paper, except for actin, are not suitable for qPCR analysis of bacterial abundance. Given that the authors used qPCR technology, it is recommended to conduct absolute quantification PCR for key bacteria (such as *Enterobacter bugandensis*, *Enterobacter roggkampii*, and *Providencia rettgeri*) with significant changes in abundance. This approach will help determine whether there are significant differences between males and females or between laboratory and field samples. Compared to the relative abundance obtained from 16S rRNA amplicon sequencing, absolute quantification PCR can more accurately reflect the actual quantity of bacteria, avoiding the influence of proportional differences in sample data on the results.

Re: We appreciate the reviewer's valuable suggestions on qPCR analysis. Currently, technical constraints prevented us from conducting species-specific absolute quantification in a short time, due to the highly conserved 16S rDNA sequences among *Enterobacter* strains and their high similarity to closely related species. To avoid the influence of proportional differences in sample data on the results, we added an absolute quantification of density of total bacteria and the key genus *Enterobacter* across different age groups and between sexes (See Fig. 1H; Fig. 3J, 3K; lines 100-102 on page 5 and lines 157-160 on page 7 of the revised manuscript).

10. Antibiotic Treatment Experimental Results: In the antibiotic treatment experiments, the authors fed newly emerged rice leaf roller moths with honey water containing three different concentrations of antibiotics, but the paper does not present the differences in results among the three antibiotic treatments nor specify which population was used after antibiotic treatment. This makes the presentation of experimental results incomplete and fails to clearly demonstrate the impact of antibiotic concentration on the results. It is suggested that the authors supplement relevant results, clarify the differences between treatment groups, and specify the populations used to help readers better understand the experimental conclusions.

Re: Sorry for this confusion. In this study, we used a single antibiotic treatment (not three concentrations) to deplete gut microbiota. We clarified the method in the revised version (See lines 357-358 on page 14).

11. Ecological Experimental Results: Some ecological experimental results in the supplementary table, such as the analysis of the feeding content of rice leaf roller moths, are interesting, but the authors did not indicate or discuss the reasons for conducting this experiment and its special biological significance, making it difficult for readers to understand the relevance of this experiment to the overall study. It is recommended that the authors provide explanations and discussions on these ecological experimental results in appropriate sections of the text to enhance the logical coherence of the research.

Re: We greatly appreciate the reviewer's constructive suggestion. We have added detailed explanations and discussions of the feeding content analysis in the main text and discussion section (see lines 180-182 on page 8 and lines 256-261 on page 11 of the revised manuscript), clarifying the experimental rationale, biological significance and relevance of this assay to the overall study.

12. Figure 5 Legend Presentation: In Figure 5, Figures 5C and 5D analyze the oviposition quantity and other indicators of insects after bacterial infection, and this

experiment is a one-way ANOVA. The current legend presentation is not easy to understand and may cause confusion. It is suggested that the authors use common labeling methods such as A, B, C, D to represent different treatment groups, making the legend clearer and easier for readers to interpret the experimental results.

Re: Thank you for the suggestion. We have revised the legend of Figure 5 to use the common labeling method (a, b, c, d) for different treatment groups, which enhances readability and interpretation (See Fig. 5D, 5G, 5J in the revised manuscript).

13. In lines 251 and 252, the authors suggest that nutritional status and gut microbiota synergistically activate the nutritional and hormonal signaling pathways in rice leaf roller moths, but lack supporting evidence for this claim, indicating an overinterpretation in the discussion. Moreover, the entire discussion section lacks clarity in logic, and the causal relationships between the evidence chains are not direct, making it difficult for readers to understand the connection between the research results and conclusions. It is recommended that the authors reorganize the discussion section content to present research findings and their significance with more rigorous logic and sufficient evidence, avoiding overinterpretation and vague causal relationship statements.

Re: We fully agree the reviewer's comments. To avoid overinterpretation and vague statements, we reorganized the discussion about the effect of nutritional status on the reproduction (lines 248-261 on page 10-11 of the revised manuscript).

In summary, it is hoped that the authors can comprehensively and thoroughly address the above issues to improve the research content and paper presentation, thereby enhancing the quality of the paper. If the authors cannot effectively resolve these issues, consideration for rejection may be necessary.

Re: We sincerely thank the reviewer for the critical and constructive comments. We have added necessary experiments, and carefully revised all the aforementioned issues to improve the paper's quality.

Reviewer #2 (Comments for the Author):

This study investigates the gut bacterial community structure and its relationship with reproduction in lab and field populations of the rice leaf folder, *Cnaphalocrocis medinalis*. The authors found that during ovarian development, bacterial load and the abundance of Enterobacteriaceae significantly increased and positively correlated with fecundity. Using antibiotic treatment (ABX) and monoassociation, the study demonstrated that gut bacteria regulate reproduction in a nutrient-context dependent manner: inhibiting it under low nutrition but showing compensatory ovarian growth under high nutrition. Key species like *Enterobacter bugandensis* and *E. roggkampii* were identified as reproductive promoters. The study addresses an important ecological question, but the manuscript requires improvements in logical consistency, citation accuracy, and mechanistic depth. Specifically, the conflict in the introduction must be resolved, and the discussion on molecular pathways (like 20E synthesis) needs to be strengthened with recent literature and, ideally, preliminary validation.

Major Comments

1. Logical Conflict in Introduction (Lines 56-58): The statement in lines 56-58 suggesting that gut microbiota have no significant effect on adult lepidopteran biology directly contradicts the following paragraph, which describes how gut bacteria promote reproduction in *Spodoptera frugiperda*. This transition is jarring. I recommend either rephrasing this section or deleting the "no effect" claim in lines 56-58 to maintain a consistent logical flow toward the role of microbes in reproduction.

Re: Thank you for your comment. We have revised the contradictory statement and rephrasing this section as “Multiple studies have focused on the microbiome in

lepidopteran larvae, but the functional roles of the gut microbiome in lepidopteran adults remains limited.” (See lines 56-57 on page 3 of the revised manuscript)

2. Molecular Mechanisms and 20E Synthesis (Lines 249-250): When discussing hormonal regulation, the authors should consider whether the gut microbiota promotes reproduction by facilitating the synthesis of 20-hydroxyecdysone (20E). I suggest adding this point to lines 249-250 and citing recent evidence (e.g., Microbiome (2025) doi.org/10.1186/s40168-025-02278-5). Furthermore, experimental validation (e.g., qPCR of Vg genes) is still highly recommended.

Re: We thank the reviewer for this constructive input. As recommended, we have added this reference (Chu et al., 2025, Microbiome, 13(1):132) in the discussion to clarify that gut microbiota may facilitate 20E synthesis to regulate reproduction (See lines 272-273 on page 11 of the revised manuscript).

Regarding experimental validation (e.g., Vg gene qPCR), we note this valuable recommendation. The present study focuses on the relationship between gut microbial community composition and host reproduction, as well as the identification of key gut symbionts in regulating *C. medinalis* reproduction. Detailed investigations into the underlying molecular mechanisms are planned for subsequent research.

3. Compensatory Development under High Nutrition: The phenomenon where ABX treatment leads to increased ovary size under 10% HS is intriguing. The authors should discuss whether this represents a fitness trade-off (e.g., at the cost of longevity) rather than just reporting the morphological change.

Re: We appreciate the insightful suggestion. As recommended, we have added references (Attisano et al., 2012, J Evol Biol, 25(5):873-80; Promislow et al., 2022, Annu Rev Entomol, 67:83-103; Ahmed et al., 2020, Nature, 584(7821):415-419.) and revised relevant discussions in the text to address this potential fitness trade-off underlying the increased ovary size in ABX-treated females fed with 10% HS (See lines 254-261 on page 11 of the revised manuscript).

4. Data on the colonization dynamics of re-introduced bacteria are needed. Additionally, survival curves for the *P. rettgeri* group must be provided to ensure that its inhibitory effect on reproduction is not simply a side effect of host morbidity/pathogenicity.

Re: Thank you for raising these important points. We have added 5 experiments to address both concerns in the revised manuscript. The results showed that *E. bugandensis*, *E. roggenkampii* and *P. rettgeri* could successfully colonize the gut of female adults (See Fig. 5B, C, F, I; lines 198-199, 204-205 and 216 on page 8-9 of the revised manuscript). In addition, our supplementary experiments found that *P. rettgeri* had no effect on the longevity of female adults, indicating that its fecundity-inhibiting effect may stem from a unique molecular mechanism rather than merely a side effect of host morbidity or pathogenicity (See Fig. 5E; lines 203 on page 9 of the revised manuscript).

Minor Comments

1. Line 70): Reference 16 cited in line 70 focuses on insecticide detoxification, which is not directly relevant to the current discussion on host reproduction. Please replace it with more appropriate references, such as doi.org/10.1038/s41522-023-00435-y and [doi: 10.1111/1749-4877.12911](https://doi.org/10.1111/1749-4877.12911).

Re: Thank you for your comment. We have replaced the less relevant Reference 16 with the suggested and more appropriate references as advised (See lines 68-69 on page 3 of the revised manuscript).

2. Word Choice (Line 64): Change "poorly understand" to "elusive."

Re: Thank you. We have changed "poorly understand" to "elusive" in Line 63 as advised.

3. Figure Clarity: Improve the visibility of scale bars in Figure 1C and increase font sizes in the Figure 2H heatmap.

Re: Following your suggestion, we have improved the visibility of the scale bars in

Figure 1C and increased the font sizes in the Figure 2H heatmap.

4. Feeding Stats: Explicitly report statistical values (e.g., P-values) in the Results to confirm that ABX treatment does not influence food intake.

Re: Revised as suggested. Statistical values (P-values) confirming that ABX treatment does not influence food intake are now explicitly reported in the Results section (See lines 183-184 on page 3 of the revised manuscript).

Re: Spectrum03878-25R1 (**Diversity of the *Cnaphalocrocis medinalis* gut bacterial community and its contribution to reproduction**)

Dear Dr. Qinjian Pan:

Your manuscript has been accepted, and I am forwarding it to the ASM production staff for publication. Your paper will first be checked to make sure all elements meet the technical requirements. ASM staff will contact you if anything needs to be revised before copyediting and production can begin. Otherwise, you will be notified when your proofs are ready to be viewed.

Sincerely,
John Chaston
Editor
Microbiology Spectrum

Reviewer #1 (Comments for the Author):

The authors have responded very carefully to the previous review comments and have supplemented the manuscript with a considerable amount of new experimental data. These additions have significantly improved the logical flow of the paper, the clarity of the results, and the depth of the discussion. As a result, the overall quality of the manuscript has been substantially enhanced. In my opinion, the paper now meets the standards required for publication.

Regarding the justification for using antibiotic treatment to assess phenotypic changes within a single generation, the authors cite several studies on agricultural pests (e.g., Zhang et al., Cell Reports, 2025, 44(7):115911; Zhang et al., ISME Journal, 2025, 19(1):wraf237; Hammer et al., 2017, PNAS, 114(36):9641-9646). However, citing prior examples does not necessarily validate the approach; logically, the mere fact that others have used this method does not automatically make it correct or appropriate for the current system. In contrast, the result showing that reintroducing symbiotic bacteria restored egg-laying capacity under low-nutrient conditions is more compelling and directly supports the functional role of the symbionts. This latter evidence carries greater weight in arguing for the biological significance of the observed phenotypes.

I recommend acceptance in its current revised form.

Reviewer #2 (Comments for the Author):

The authors have successfully addressed my previous concerns. I have no further comments.